# Gonadotropin-Releasing Hormone agonist (GnRH-a) Pretreatment before Hormone Replacement Therapy Does Not Improve Reproductive Outcomes of Frozen–Thawed Embryo Transfer Cycle in Older Patients with Intrauterine Fibroid: A Retrospective Cohort Study

**DOI:** 10.3390/jcm12041401

**Published:** 2023-02-10

**Authors:** Ping Sun, Yanlei Dong, Yi Yu, Hui Xu, Lin Zhu, Ping Zhang, Lei Yan

**Affiliations:** 1Center for Reproductive Medicine, Shandong University, Jinan 250033, China; 2Department of Gynecology, The Second Hospital of Shandong University, Jinan 250033, China

**Keywords:** endometrial preparations, frozen-thawed embryo transfer, live birth rate, clinical pregnancy rate, geriatric patients, intermural fibroid

## Abstract

Background: Surgery in elder patients with intermural fibroids delays pregnancy, and GnRH-a can shrink uterine fibroids to a certain extent; therefore, for geriatric patients with fibroids, determining whether GnRH-a pretreatment before frozen–thawed embryo transfer (FET) can improve its success rate remains to be studied. We conducted this study to research whether GnRH-a pretreatment before hormone replacement treatment (HRT) could optimize the reproductive outcomes compared with others preparations in geriatric patients with intramural fibroids. Methods: According to the endometrial preparation, patients were divided into a GnRH-a–HRT group, a HRT group and a natural cycle (NC) group. The live birth rate (LBR) was the first outcome, and the clinical pregnancy outcome (CPR), the miscarriage rate, the first trimester abortion rate and the ectopic pregnancy rate were the secondary outcomes. Results: A total of 769 patients (aged 35 years or older) were included in this study. No significant difference was observed in the live birth rate (25.3% vs. 17.4% vs. 23.5%, *p* = 0.200) and the clinical pregnancy rate (46.3% vs. 46.1% vs. 55.4%, *p* = 0.052) among the three endometrial preparation regimens. Conclusion: In this study, for the geriatric patient with the intramural myoma, the pretreatment with GnRH-a did not show any advantage over the NC and HRT preparation groups before the FET, and the LBR was not significantly increased.

## 1. Introduction

With the developments in embryo cryopreservation techniques in recent decades, frozen embryo transfer (FET) has played an essential role in Assisted Reproductive Techniques (ART) owing to its many advantages. Apparently, FET can reduce the number of times ovarian stimulation and ovarian puncture is repeated if fresh embryo transfer is unsuccessful and extra embryos are available [1,2,3]. The exact synchronization between embryo development and endometrial maturation plays a crucial role in the success of frozen embryo transfer. In order to improve the success of implantation, various endometrial preparation protocols have been explored. FET preparation methods can largely be divided into NCs and artificial cycles, also called HRTs [4,5]. GnRH-a can be used before HRT, and could down-regulate the pituitary function. Up to now, GnRH-a down-regulation is usually applied to women with recurrent implantation failure, endometriosis and adenomyosis, and many researches have proved that GnRH-a suppression significantly enhances the chances of pregnancy [6,7,8,9]. For patients with uterine fibroids, GnRH-a is mainly used to improve preoperative symptoms and reduce the size of fibroids, as well as to prevent the recurrence of uterine fibroids after surgery [10,11]. The mechanism of the GnRH-a to reduce fibroid volume is thought to reduce the expression of the basic fibroblast growth factor and exhibit a negative effect on various signal transduction pathways stimulated by gonadal hormones [12]. However, at present, there is a lack of studies on endometrial preparation regimens for uterine fibroids, and there are no reports on whether GnRH-a pretreatment programs will affect the pregnancy outcome of uterine fibroids.

Regarding intramural fibroids, their effect on patient’s reproductive outcomes remains controversial. Extensive research has reported that intramural fibroids, even including non-cavity-distorting intramural fibroids, effect reproductive outcomes [13,14,15,16]. A meta-analysis, which was in regard to the effect of intramural fibroids on the outcome of IVF-ET, showed that in addition to lowering clinical pregnancy rates and live birth rates, the rate of miscarriage was significantly increased in the fibroid group [17]. A 2018 retrospective cohort study conducted by Lei Yan et al. found that type 3 intermural myoma had significant adverse effects on the implantation rate, clinical pregnancy rate, and live birth rate in IVF/ICSI cycles [18]. Therefore, these researchers believe that intramural myoma has an impact on the outcome of IVF/ICSI, and surgical intervention in advance may be a feasible approach. However, well-designed surgical intervention trials for myomectomy and infertility are sparse, and the possible complications that caused by myomectomy are worth serious consideration; these include, for example, infection, damage to internal organs, and risk of blood or blood product transfusions [19,20]. For geriatric patients, in addition to the complications that may be caused by the surgery described above, an awkward problem that cannot be ignored is that it takes at least six months to a year to recover from pregnancy after surgery, in order to reduce and avoid the risk of uterine rupture after pregnancy. However, during the waiting period, uterine fibroids may recur and the success rate of in vitro pregnancy assistance may be affected with the increase in age.

A multitude of researchers have conducted various research to compare the application of different endometrial preparation schemes in the population. There is no sufficient evident to support one preparation in preference to another for FET in women with regular ovulatory cycle [3]. The latest review analyzed seven articles and showed that patients treated with HRT plus GnRH-a suppression achieved higher LBR than women treated with HRT alone [4]. It is suggested that, in order to not increase the cost of patients and GnRH-a side effects, women with normal menstrual cycles do not need prior pituitary down-regulation before HRT. In polycystic ovary syndrome (PCOS) patients, GnRH-a–HRT also cannot improve the LBR compared with the HRT cycle [21]. Nevertheless, there are few relevant studies on the comparison of endometrial preparation schemes in patients with intrauterine fibroids, especially in geriatric patients.

Based on this, we considered whether there would be a more suitable endometrial preparation regime for patients over the age of 35 years with uterine fibroids during frozen embryo transplantation. Because it is hard for these patients to choose surgery or embryo transfer, we wonder if the GnRH-a–HRT would be more advantageous in these patients. We conducted this retrospective cohort study to compare the three endometrial preparation regimens in a population with intramural fibroids to explore if the GnRH-a–HRT regime has an advantage in such patients.

## 2. Materials and Methods

### 2.1. Study Design and Participants

In this retrospective cohort study, medical records were extracted and retrospectively reviewed for women who underwent frozen embryo transfer between February 2017 and December 2021 in the Reproductive Hospital Affiliated to Shandong University. The frozen embryos of all the patients were available after their first ovulation collection cycle. The inclusion criteria were as follows: (1)age ≥ 35 years old; (2) patients diagnosed with an intramural myoma by ultrasound; and (3) only NCs, HRTs and GnRH-a–HRT endometrial preparations. The exclusion criteria were as follows: (1) patients with endometriosis or adenomyosis; (2) untreated patients with submucous fibroids, hydrosalpinx, endometrial polyps, uterine adhesions and uterine malformations affecting the success rate of pregnancy; (3) patients who underwent embryo transfer after PGT or those suffering from chromosomal disease; (4) patients with recurrent abortion and recurrent implantation failure; (5) patients who receive sperm or egg donations; and (6) patients with PCOS. The research protocol was approved by the hospital institutional ethics committee (No. 2022-58).

### 2.2. Endometrial Preparations

#### 2.2.1. Natural Cycles

Patients who underwent the NC endometrium preparation regimen began to monitor follicular growth and the endometrium by transvaginal ultrasound from the 10th to 12th day of menstrual cycle until ovulation or luteinization, and blood or urine LH detection could be used to assist diagnosis if necessary. Given that there are individual differences in the timing of luteal peaks, we could also intramuscularly inject 6000 to 8000 IU of human chorionic gonadotropin (HCG) when the leading follicles were developing to 17–18 mm in order to stimulate the formation of endogenous luteal peaks and promote follicle maturation [22]. The ovulation day was set as D0, and cleavage-stage embryo transplantation was performed on the 3rd day after ovulation; blastocyst transplantation was performed on the 5th day after ovulation. Progesterone was added from the ovulation day or the first day after ovulation.

#### 2.2.2. Hormone Replacement Treatment Cycles

In the HRT cycles, 4–6 mg of estradiol valerate or 17β estradiol was given to the patients from the 2nd to 3rd day of menstruation daily, with the purpose of promoting endometrial proliferation and inhibiting the appearance of leading follicles. On the 10th to 12th day of medication, transvaginal ultrasound was performed to check the endometrial thickness. When the endometrial thickness reached at least 7 mm, progesterone was added to achieve endometrial transformation [22]. Embryo transfer was performed five to seven days later, according to the physician’s discretion.

#### 2.2.3. Hormonal Replacement Treatment with GnRH agonist Pretreatment (GnRH-a–HRT)

In the GnRH-a–HRT regimen, on days 1–4 of menstrual period, patients were injected with 3.75 mg of GnRH-a to down-regulate the function of the pituitary. After about 30 days, we evaluated whether patients had reached the pituitary down-regulation status by evaluating ultrasound and hormone levels. The standard criteria for the pituitary down-regulation are as follows: luteinizing hormone (LH) < 5 IU/L, uterine endometrial thickness <5 mm, estrogen (E2) < 50 pg/mL and no large follicles or cysts. When the standard criteria were reached, hormone replacement followed, as described in the HRT regimen.

### 2.3. Outcome Assessment

The primary outcomes are LBR and CPR in this retrospective research. LBR is defined as the number of deliveries that resulted in at least one live birth, expressed per 100 embryo transfer cycle attempts [23]. The clinical pregnancy means a pregnancy diagnosed by the ultrasonographic visualization of gestational sacs; in addition to the intrauterine pregnancy sac, ectopic pregnancy also belongs to clinical pregnancy. The clinical pregnancy rate is defined as the number of clinical pregnancies expressed per 100 embryo transfer cycles. The second outcomes are miscarriage rate, ectopic pregnancy rate and first trimester miscarriage rate. The miscarriage rate is defined as the loss of clinical pregnancy before the 28th gestational week, divided by the total number of clinical pregnancy cycles. The first trimester miscarriage means spontaneous abortion within 12 weeks of gestation after confirmation of pregnancy, and the first trimester miscarriage rate is calculated as the number of first trimester miscarriages divided by the number of clinical pregnancy cycles. The ectopic pregnancy rate is calculated by the number of ectopic pregnancy cycles, divided by the number of clinical pregnancy cycles.

### 2.4. Statistical Analysis

IBM SPSS Statistics 25 (v. 25.0; International Business Machines Co., Armonk, NY, USA) was used for statistical analysis. We first used the multiple interpolation method to deal with the missing values. We conducted the normal distribution test for continuous data. If the data met the normality, the t test was used, and the results were presented by mean ± standard deviation. If the normal distribution was not followed, the Kruskal–Wallis H test was used, and the results were expressed as median and quartiles. The Chi-square test was used for categorical variables, and percentage (number) was used for statistical description. Since a patient could contribute data for multiple frozen embryo transfer cycles, we used the generalized estimation equation (GEE), which is based on logistic regression to control the confounding factors. A *p*-value under 0.05 was considered to be significant.

## 3. Results

This retrospective study finally included 769 patients and they were divided into three groups:

The GnRH-a–HRT group (*n* = 95), the HRT group (*n* = 167) and the NC group (*n* = 507). The demographic and baseline characteristics of the participants are presented in Table 1.

### 3.1. Baseline Characteristics and Reproductive Outcome

According to the results shown in Table 1, significant differences in the body mass index (BMI) are observed between the HRT group and the NC group (25.39 (23.26~27.98) vs. 23.94 (21.84~26.50), *p* < 0.001); The Anti-Mullerian Hormone (1.76 (1.23~3.23) vs. 2.56 (1.28~4.67), *p* = 0.034) and basal follicle-stimulating hormone (6.72 (5.89~8.46) vs. 6.25 (5.44~7.43), *p* = 0.022) are significantly different between the GnRH-a–HRT group and the HRT group. Significant differences were also observed in antral follicle counting (AFC) (11 (7.5~15) vs. 12 (8~20) vs. 12 (8~15), *p* = 0.032), endometrial thickness (1 (0.85~1.1) vs. 0.85 (0.8~1) vs. 0.95 (0.83~1.1), *p* < 0.001) and maximum fibroid diameter (2.3 (1.70~3.25) vs. 1.9 (1.34~2.60) vs. 2.0 (1.40~2.60), *p* = 0.004). There was no statistical difference in endometrial thickness between the GnRH-a–HRT group and the NC group, but there were statistical differences between the two groups and the HRT group (1 (0.85~1.1) vs. 0.85 (0.8~1) vs. 0.95 (0.83~1.1), *p* < 0.001), and we could observe that the endometrial thickness in the GnRH-a–HRT group and the NC group was thicker than that in the HRT group. In terms of the maximum fibroid diameter, there was no significant statistical difference between the HRT group and the NC group, but the fibroid diameters of these two groups were smaller than the GnRH-a–HRT group and had a statistical difference (2.3 (1.70~3.25) vs. 1.9 (1.34~2.60) vs. 2.0 (1.40~2.60), *p* = 0.004). As the results show in Table 2, the differences in the live birth rate (25.3% (24) vs. 17.4% (29) vs. 23.5% (119), *p* = 0.200), the clinical pregnancy rate (46.3% (44) vs. 46.1% (77) vs. 55.4% (281), *p* = 0.052), the miscarriage rate (36.4% (16) vs. 45.5% (35) vs. 42.2% (118), *p* = 0.619), the first trimester miscarriage rate (18.2% (8) vs. 23.4% (18) vs. 20.3% (57), *p* = 0.789) and the ectopic pregnancy rate (0.0% (0) vs. 0.0% (0) vs. 1.1% (3), *p* = 0.718) were all inconspicuous among these three endometrial preparations.

### 3.2. GEE Analysis of Factors Related to LBR and CPR

A generalized estimation equation was produced using the live birth rate and the clinical pregnancy rate as dependent variables, and age, age at CET, BMI, AFC, AMH, basal FSH, types of infertility, methods of ART, endometrial thickness on luteal support day, the number of embryos transferred, maximum fibroid diameter, number of fibroids, and endometrial preparation as independent variables. The results presented in Table 3 showed that BMI and AFC can significantly influence both the live birth rate and the clinical pregnancy rate; in addition to this, the endometrial thickness on the luteal support day is also an important factor that could efficiently affect the live birth rate. Compared with a single fibroid, two fibroids and AMH could significantly influence the clinical pregnancy rate. The other factors have no effect on the live birth rate and the clinical pregnancy rate.

Subgroup analysis was performed for patients with uterine fibroids ≥3 cm in diameter; the results presented in Table 4 show that the endometrial thickness on luteal support day is significantly different between the GnRH-a–HRT group and the HRT group (1 (0.9~1.2) vs. 0.88 (0.83~1) s 0.93 (0.85~1.1), *p* = 0.028). Statistical difference can be observed in the maximum fibroid diameter between the HRT group and the NC group (4.55 (3.31~5.9) vs. 3.6 (3.2~4.4), *p* = 0.006). There is no difference among the three endometrial preparations in terms of the live birth rate and clinical rate.

## 4. Discussion

In this retrospective cohort study, we found that there was no significant difference in the live birth rate, clinical pregnancy rate, abortion rate, first trimester abortion rate and ectopic pregnancy rate among three endometrial preparation schemes, but the maximum diameter of intrauterine fibroids in GnRH-a–HRT group was larger than that in the other two groups. After the application of the GnRH-a–HRT regimen, the endometrial thickness was improved and it was thicker than that in the HRT group, with statistical difference. The results indicated that the GnRH-a–HRT regimen could optimize endometrial thickness, suggesting that preconditioning may be beneficial for people with a thin endometrium. The same was true for patients with intrauterine fibroid ≥3 cm. However, for geriatric patients with intrauterine fibroid infertility, the GnRH-a–HRT regimen showed no significant advantage in improving the live birth rate and clinical pregnancy rate. BMI, AFC and endometrial thickness are mean factors influencing the live birth rate.

To date, many researchers have analyzed and compared the reproductive outcomes in different populations using distinct endometrial regimens [21,24,25,26]. To our knowledge, we are the first study to compare the use of three endometrial preparation regimens in older patients with intrauterine fibroid. As we all know, the key to the successful pregnancy of freeze–thaw embryo transfer is the synchronization of endometrium and embryo development, the premise of which is to have a good endometrium planting environment [4]. Several studies report a positive effect on pregnancy rates of a thickened endometrium in FET [27,28,29]. A prospective randomized study that included 106 patients compared the pregnancy outcome with or without prior gonadotrophin-releasing hormone agonist suppression, found a similar pregnancy rate, implantation rate and ongoing pregnancy rate; no significant difference was found in endometrial thickness [30]. Another retrospective cohort study, stratified by the number of times embryo implantation failed, demonstrated that the live birth rate was raised in patients with multiple embryo implantation failures in the GnRH-a–HRT group [8]. They also found that GnRH-a pretreatment can ameliorate endometrial thickness on progesterone initiation day independently from the number of implantation failures. Our study included patients who underwent a maximum of four frozen embryo transfer cycles, including periods without pregnancy after transplantation, and we excluded patients with recurrent miscarriage. The result of our research is in agreement with the latter research, demonstrating that GnRH-a pretreatment also could improve endometrial thickness in older patients with intrauterine fibroid. However, no significant difference was observed in reproductive outcomes, which may be attributed to the limited number of participants and the large difference in the population size between the groups. A large prospective cohort study or randomized controlled study is urgently needed to confirm whether a GnRH-a preconditioning regimen will improve pregnancy outcomes in this population by improving endometrial thickness.

According to the existing literature, several main mechanisms that control how uterine fibroids affect the endometrium and reproductive outcome are reported, including uterine cavity distortion, increased uterine contractility, impaired endometrial and myometrial blood supply, impaired endometrial receptivity and gene expression, and a thicker capsule [31]. Uterine cavity distortion is considered to be one of the most significant mechanisms involved [32,33]. By magnetic resonance imaging studies, researchers found that the blood supply to the myometrium surrounding uterine fibroids decreased [34]. In addition, the blood flow velocity of uterine fibroids increased under transvaginal ultrasound, and the resistance index decreased; in addition, the uterine artery pulsation index decreased. Therefore, the presence of uterine fibroids may affect the implantation of embryos by causing changes in the blood supply [35,36]. Intramural myomas caused abnormal uterine peristalsis, resulting in lower implantation and pregnancy rates [35,37]. The successful implantation of embryos depends on a series of processes, such as localization, adhesion and invasion. The presence of uterine fibroids could lead to the decreased expression of proteins and molecules related to embryo implantation. In 2010, Ben-Nagi et al. found decreased levels of glycodelin and interleukin (IL)-10 in uterine flushing fluid of the endometrium during the mid-luteal phase in women with fibroids [36,38]. IL-11 and E-cadherin, which play important role in regulation of trophoblast invasion and adhesion, were also found to be decreased in expression. Compared to the control group, HOXA-10, which is an important gene that regulates endometrial receptivity, was impacted in the intramural group [31]. Currently, a multitude of studies reported that intramural fibroid could cause infertility and significantly affect the reproductive outcome in patients undergoing IVF/ICSI treatment [14,18,39,40]. A recent systematic review and meta-analysis including 15 studies with 5029 patients reported that, compared to women with no fibroids, patients with non-cavity-distorting intermural fibroids had 44% lower odds of live birth and 32% lower odds of clinical pregnancy. Subgroup analysis of patients with only intramural fibroids found critically lower odds of live birth rates and clinical pregnancy rates. Analysis of prospective and retrospective studies presented that even non-cavity-distorting intramural fibroids certainly have a detrimental effect on live birth rates in patients undergoing IVF treatment [14].

At present, it is still controversial as to whether to perform surgical treatment before IVF for intrauterine myoma without uterine cavity deformation, especially for the geriatric patients; intraoperative or postoperative complications, and the time required to wait for a second pregnancy after surgery are all factors and costs that cannot be ignored [40,41,42]. A number of studies have discussed the role of GnRH-a in reducing the uterus and fibroid volume before myomectomy, as well as delaying the recurrence of multiple uterine fibroids after surgery [10,11,43]. In GnRH-a–HRT endometrial preparation cycles, although GnRH-a is used to down-regulate the function of the pituitary, it is unknown whether the application of GnRH-a could improve the pregnancy outcome in these older patients with intrauterine fibroids. The efficacy of GnRH-a pretreatment before hormonal replacement treatment is also controversial [8]. Based on the above considerations, we conducted this retrospective cohort study to explore whether GnRH-a pretreatment could optimize the pregnancy outcome compared with the NC group and the HRT group. At present, GnRH-a down-regulation programs are commonly used in patients with endometriosis, adenomyosis, polycystic ovarian syndrome, patients with decreased ovarian reserve, and patients with repeated implantation failure. In order to exclude other factors that might affect the endometrium, we excluded patients with adenomyosis, endometriosis, PCOS, recurrent abortion, and uterine abnormalities. In view of the clustering effect caused by multiple frozen embryo transfer cycles for each patient, GEE analysis was used to replace the traditional binary logistic regression analysis. According to the GEE analysis, the type of endometrial preparation is not factor significantly influencing the live birth rate and clinical pregnancy rate; however, the intima thickness is, and in the GnRH-a–HRT group, it has been significantly optimized. Multiple studies have shown that the endometrial thickness is correlated with the outcome of ART pregnancy [8,30,44]. Endometrial thickening can increase the duration of the pregnancy, the possibility of pregnancy and live birth, and is independent of the effects of age and embryo quality [27,28,45]. Although the specific mechanisms are still unclear, it may be due to the interruption of the continuous menstrual cycle caused by the prolonged down-regulation of the pituitary gland, which may restore the full function of the hormone-sensitive system [46].

Our present study has several advantages. To our knowledge, no consensus was found on which preparation regimen could lead to a better reproductive outcome for older patients with intermural fibroids. The results of our study can provide some reference for daily clinical work. Secondly, we formulated strict inclusion and exclusion criteria when screening the study population, excluding diseases that may affect the reproductive outcome and the status of the uterine cavity and endometrium, such as endometriosis, adenomyosis, uterine malformation, etc. The patients that were diagnosed with recurrent spontaneous abortion and recurrent implantation failure were also excluded. Thirdly, there were no strict restrictions on exogenous hormone administration and dosage forms, depending on the clinician’s habits and patient preferences. A multitude of research has reported that the outcomes of FET are comparable to any mode of administration or dosage form [47]. Last but not least, we used appropriate data analysis methods, and GEE analysis was used instead of the conventional logistic regression analysis, given the possibility of multiple FET cycles per patient.

There are also some disadvantages in this study. First of all, this is a retrospective cohort study with inherent limitations, and we were unable to investigate other confounding factors such as exercise, diet and so on. In addition, there was a significant difference in the number of patients in the three endometrial preparation groups; however, this depends on the preference and experience of clinicians when choosing preparation treatment, so this is a bias that we cannot control and avoid. Furthermore, all of the FET cycles included in our study were carried out after the first ovulation cycle; although GEE analysis was performed to correct confounding factors, bias-related factors due to embryo selection could not be avoided.

## 5. Conclusions

In conclusion, our research suggests that GnRH-a pretreatment before HRT cannot reduce the size of intramural fibroids through down-regulation in order to improve reproductive outcomes. Compared with the NC cycles and HRT cycles, no better reproductive outcomes were observed in the GnRH-a–HRT group. However, the GnRH-a–HRT cycles significantly improved the endometrial thickness; this may have implications for patients with a thin endometrium. Due to the limitations of retrospective studies, a further randomized controlled trial or a larger prospective cohort study needs to be conducted to confirm this conclusion and explore the effect of down-regulation duration on the pregnancy outcomes of fibroid patients. The comparison between the long-term down-regulation and the surgical resection of fibroids also need to be discussed.

## Figures and Tables

**Table 1 jcm-12-01401-t001:** Demographic and characteristics of patients among three endometrial preparation regimens.

	GnRH-a–HRT Group (*n* = 95)	HRT Group (*n* = 167)	NC Group (*n* = 507)	*p*-Value
Age	38(36~40)	38(36~39)	38(36~39)	0.739
Women’s age at CET	38(37~40)	37(36~39)	38(36~39)	0.073
BMI	24.88(22.75~27.27)	25.39(23.26~27.98) ^a^	23.94(21.84~26.50) ^b^	<0.001 *
AMH	1.76(1.23~3.23) ^a^	2.56(1.28~4.67) ^b^	2.22(1.21~3.62)	0.034 *
AFC	11(7.5~15)	12(8~20)	12(8~15)	0.032 *
Basal FSH	6.72(5.89~8.46) ^a^	6.25(5.44~7.43) ^b^	6.76(5.6~7.88)	0.022 *
Basal LH	4.49(3.35~6.34)	4.3(3.41~6.15)	4.52(3.55~5.71)	0.854
Basal E2	39.1(27.6~49.6)	37.8(27.15~48.94)	37.7(28.1~49.65)	0.915
Basal *p*	0.3(0.16~0.47)	0.25(0.17~0.43)	0.28(0.17~0.46)	0.541
Basal PRL	15.59(10.58~18.89)	14.3(10.53~19.28)	15.16(10.94~19.41)	0.431
Duration of infertility (years)	3.5(2~5.5)	3(1.75~5)	3(2~6)	0.654
Types of infertility				0.713
Primary infertility	20.0%(19)	19.8%(33)	22.5%(114)	
Secondary infertility	80.0%(76)	80.2%(134)	77.5%(393)	
IVF/ICSI				0.121
IVF	82.1%(78)	71.9%(120)	78.1%(396)	
ICSI	17.9%(17)	28.1%(47)	21.9%(111)	
Number of oocyte retrieval	9(6~13)	11(7~16)	10(6~14)	0.051
2PN embryos	7(4~10)	7(4~11)	7(4~9)	0.156
Endometrial thickness	1(0.85~1.1) ^b^	0.85(0.8~1) ^a^	0.95(0.83~1.1) ^b^	<0.001 *
Number of embryos transferred	1(1~1)	1(1~1)	1(1~1)	0.276
Blastocyst transfer rate	100%(95)	99.4%(166)	99.6%(505)	1.000
High-quality embryonic rate	77.9%(74)	82.6%(138)	83.6%(424)	0.407
Maximum fibroid diameter	2.3(1.70~3.25) ^a^	1.9(1.34~2.60) ^b^	2.0(1.40~2.60) ^b^	0.004 *
The number of fibroids				0.093
1	55.8%(53) ^a^	71.3%(119) ^b^	63.9%(324)	
2	8.4%(8)	8.4%(14)	9.7%(49)	
≥3	35.8%(34) ^a^	20.4%(34) ^b^	26.4%(134)	

CET: Cyro-embryo transfer; BMI: Body Mass Index FSH: Follicle-Stimulating Hormone; AMH: Anti-Mullerian Hormone; AFC: Antral follicle counting; LH: Luteinizing hormone; E2: Estradiol; PRL: Prolactin; PRL: Prolactin; IVF: In vitro fertilization; ICSI: Intracytoplasmic sperm injection; PN: Pronucleus; GnRH-a: Gonadotropin releasing hormone agonist; HRT: Hormone replacement treatment; NC: natural cycle; a/b: If they are marked with the same symbol, there is no statistical difference between groups; if they are different, there is a statistical difference. *: *p* < 0.05.

**Table 2 jcm-12-01401-t002:** Reproductive outcomes of three endometrial preparations.

	GnRH-a–HRT Group (*n* = 95)	HRT Group (*n* = 167)	NC Group (*n* = 507)	*p*-Value
LBR	25.3%(24)	17.4%(29)	23.5%(119)	0.200
CPR	46.3%(44)	46.1%(77)	55.4%(281)	0.052
Miscarriage Rate	36.4%(16)	45.5%(35)	42.2%(118)	0.619
First trimester Miscarriage Rate	18.2%(8)	23.4%(18)	20.3%(57)	0.789
Ectopic Rate	0.0%(0)	0.0%(0)	1.1%(3)	0.718

GnRH-a: Gonadotropin releasing hormone agonist; HRT: Hormone replacement treatment; NC: Natural cycle; LBR: Live Birth Rate; CPR: Clinical Pregnancy Rate.

**Table 3 jcm-12-01401-t003:** GEE analysis of live birth rate.

Reproductive Outcome		Live Birth Rate	
Value		OR(95%CI)	*p*
Age		1.06(0.86~1.30)	0.587
Women’s age at CET		0.86(0.71~1.04)	0.118
BMI		0.90(0.85~0.96)	<0.001 *
AMH		0.93(0.84~1.02)	0.110
AFC		1.04(1.00~1.08)	0.035 *
Basal FSH		0.99(0.91~1.08)	0.810
Types of infertility		1.24(0.81~1.89)	0.330
IVF/ICSI		1.18(0.75~1.85)	0.472
Endometrial thickness		2.82(1.02~7.66)	0.042 *
Number of embryos transferred		1.97(0.72~5.37)	0.187
Maximum fibroid diameter		0.94(0.80~1.13)	0.536
The number of fibroids	1	1	
	2	1.69(0.96~2.97)	0.067
	≥3	1.12(0.73~1.72)	0.591
Endometrial preparation regimens	GnRH-a–HRT cycle	1	-
	HRT cycle	1.69(0.96~2.97)	0.067
	NC cycle	1.12(0.73~1.72)	0.591

CET: Cyro-embryo transfer; BMI: Body Mass Index FSH: Follicle-Stimulating Hormone; AMH: Anti-Mullerian Hormone; AFC: Antral follicle counting; IVF: In vitro fertilization; ICSI: Intracytoplasmic sperm injection; GnRH-a: Gonadotropin releasing hormone agonist; HRT: Hormone replacement treatment; NC: Natural cycle; GEE: Generalized estimation equation; OR: Odds ratio; CI: Confidence interval; *: *p* < 0.05.

**Table 4 jcm-12-01401-t004:** Demographic, characteristics and reproductive outcome of patients with fibroids ≥3.0 cm among the three groups.

	GnRH-a–HRT Group (*n* = 26)	HRT Group (*n* = 28)	NC Group (*n* = 86)	*p*-Value
Age	38(37~40)	38.5(36~41.5)	38(36~39)	0.356
Women’s age at CET	38.5(37~41)	39(36~41.5)	38(36~39)	0.109
BMI	25.98(22.72~28.73)	25.82(23.36~28.19)	24.16(21.99~26.84)	0.242
AMH	1.62(1.2~3.01)	2.67(1.42~5.12)	2.33(1.24~3.38)	0.189
AFC	11(7~16)	13(9.5~17.5)	12(9~15)	0.610
Basal FSH	6.26(5.02~7.17)	6.29(5.27~7.77)	6.52(5.49~7.84)	0.416
Basal LH	4.08(3.29~5.38)	4.61(3.69~6.59)	4.57(3.68~5.4)	0.489
Basal E2	41.09(23~57.2)	38.15(27.6~45)	36.6(26.3~44.3)	0.718
Basal P	0.26(0.14~0.4)	0.23(0.15~0.48)	0.28(0.16~0.46)	0.562
Basal PRL	14.26(9.9~18.63)	12.92(10.49~17.53)	15.81(12.4~19.08)	0.171
Duration of infertility (years)	2(1~5)	2(1~5.25)	3.5(2~6)	0.136
Types of infertility				0.325
Primary infertility	11.5%(3)	25.0%(7)	14.0%(12)	
Secondary infertility	88.5%(23)	75.0%(21)	86.0%(74)	
IVF/ICSI				0.467
IVF	88.5%(23)	78.6%(22)	76.7%(66)	
ICSI	11.5%(3)	21.4%(6)	23.3%(20)	
Number of oocyte retrieval	9(5~12)	10.5(7~17)	9(6~14)	0.222
2PN embryos	6(3~9)	7.5(5~11.5)	6(4~9)	0.174
Endometrial thickness	1(0.9~1.2) ^a^	0.88(0.83~1) ^b^	0.93(0.85~1.1)	0.028 *
Blastocyst transfer rate	1(1~1)	1(1~1)	1(1~1)	0.112
High-quality embryonic rate	92.3%(24)	85.7%(24)	87.2%(75)	0.772
Maximum fibroid diameter	4.55(3.31~5.9) ^a^	4(3.45~5.1)	3.6(3.2~4.4) ^b^	0.006 *
The number of fibroid				0.058
1	38.5%(10)	64.3%(18)	44.2%(38)	
2	0.0%(0)	10.7%(3)	9.3%(8)	
≥3	61.5%(16)	25%(7)	46.5%(40)	
LBR	23.1%(6)	14.3%(4)	18.6%(16)	0.747
CPR	46.2%(12)	42.9%(12)	50.0%(43)	0.791

CET: Cyro-embryo transfer; BMI: Body Mass Index FSH: Follicle-Stimulating Hormone; AMH: Anti-Mullerian Hormone; AFC: Antral follicle counting; LH: Luteinizing hormone; E2: Estradiol; PRL: Prolactin; PRL: Prolactin; IVF: In vitro fertilization; ICSI: Intracytoplasmic sperm injection; PN: Pronucleus; GnRH-a: Gonadotropin releasing hormone agonist; HRT: Hormone replacement treatment; NC: Natural cycle; LBR: Live birth rate; CPR: Clinical pregnancy rate; a/b: If they are marked with the same symbol, there is no statistical difference between groups; if they are different, there is a statistical difference. *: *p* < 0.05.

## Data Availability

Please contact author for data requests.

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
