# Peer review of "Gonadotropin-Releasing Hormone agonist (GnRH-a) Pretreatment before Hormone Replacement Therapy Does Not Improve Reproductive Outcomes of Frozen–Thawed Embryo Transfer Cycle in Older Patients with Intrauterine Fibroid: A Retrospective Cohort Study"

_jcm, 2023, doi:10.3390/jcm12041401_

Round 1
Reviewer 1 Report
I appreciate the opportunity to review the manuscript entitled “Gonadotropin-releasing hormone agonist (GnRH-a) pretreatment before hormone replacement therapy doesn’t improve reproductive outcomes of frozen-thawed embryo transfer cycle in older patients with intrauterine fibroid: A retrospective cohort study.” submitted to the Journal of Clinical Medicine.
The authors investigated the influence of the different pretreatment protocols on outcomes of FET in patients with intramural uterine fibroids.
Specific comments:
1) Please insert the list of abbreviations used in the manuscript.
2) Please insert in the discussion section the informations about different molecular mechanisms of the interplay of uterine fibroids and endometrium, as well as these implications on the outcomes of ART techniques, especially FET.
Taking into account the importance of the topic studies in the above-mentioned paper, as well as the results that which authors observed, my opinion is that this submission meets the criteria to be published in the Journal of Clinical Medicine after the minor revision I suggested.
Author Response
Point 1: Please insert the list of abbreviations used in the manuscript.
Response 1:Thank you for your valuable comment,we have added “abbreciations” before the “Introduction” section and you can find it in line 32-52 in the latest manuscript we submitted .
Point 2: Please insert in the discussion section the informations about different molecular mechanisms of the interplay of uterine fibroids and endometrium, as well as these implications on the outcomes of ART techniques, especially FET.
Response2: Thank you very much for your constructive recommendation. We reviewed the relevant literature and discussed the relevant mechanisms and effects on IVF outcomes in the discussion section. We have highlight this section in yellow and you can find it in line 289-317. We sincerely thank you for your kind recommendation again, which have helped improve our manuscript.
Reviewer 2 Report
Fellow colleagues, congratulations on your work. As a practitioner in IVF myself, I am in agreement with your final remark, that GnRH-as a pretreatment before HRT can't result a better reproductive outcome compared with NC cycles and HRT cycles but can improve endometrial thickness in patients with chronically thin endometrium. However, I consider the conclusions to be too short and need more elaborating.
Author Response
Point 1: Fellow colleagues, congratulations on your work. As a practitioner in IVF myself, I am in agreement with your final remark, that GnRH-as a pretreatment before HRT can't result a better reproductive outcome compared with NC cycles and HRT cycles but can improve endometrial thickness in patients with chronically thin endometrium. However, I consider the conclusions to be too short and need more elaborating.
Response: Thank you very much for the kind comment, We re-edit the conclusion of the text and highlight the newly added part in yellow, you can find it in line 371-380 in revised manuscript. Thank you again for your suggestion.